# Tumor-Infiltrating Leukocyte Composition and Prognostic Power in Hepatitis B- and Hepatitis C-Related Hepatocellular Carcinomas

**DOI:** 10.3390/genes10080630

**Published:** 2019-08-20

**Authors:** Yi-Wen Hsiao, Lu-Ting Chiu, Ching-Hsuan Chen, Wei-Liang Shih, Tzu-Pin Lu

**Affiliations:** Institute of Epidemiology and Preventive Medicine, Department of Public Health, National Taiwan University, Taipei City 10617, Taiwan

**Keywords:** hepatocellular carcinoma, hepatitis B virus, hepatitis C virus, tumor-infiltrating lymphocytes, immune cell, ESTIMATE, CIBERSORT

## Abstract

**Background:** Tumor-infiltrating leukocytes (TILs) are immune cells surrounding tumor cells, and several studies have shown that TILs are potential survival predictors in different cancers. However, few studies have dissected the differences between hepatitis B- and hepatitis C-related hepatocellular carcinoma (HBV−HCC and HCV−HCC). Therefore, we aimed to determine whether the abundance and composition of TILs are potential predictors for survival outcomes in HCC and which TILs are the most significant predictors. **Methods:** Two bioinformatics algorithms, ESTIMATE and CIBERSORT, were utilized to analyze the gene expression profiles from 6 datasets, from which the abundance of corresponding TILs was inferred. The ESTIMATE algorithm examined the overall abundance of TILs, whereas the CIBERSORT algorithm reported the relative abundance of 22 different TILs. Both HBV−HCC and HCV−HCC were analyzed. **Results:** The results indicated that the total abundance of TILs was higher in non-tumor tissue regardless of the HCC type. Alternatively, the specific TILs associated with overall survival (OS) and recurrence-free survival (RFS) varied between subtypes. For example, in HBV−HCC, plasma cells (hazard ratio [HR] = 1.05; 95% CI 1.00–1.10; *p* = 0.034) and activated dendritic cells (HR = 1.08; 95% CI 1.01–1.17; *p* = 0.03) were significantly associated with OS, whereas in HCV−HCC, monocytes (HR = 1.21) were significantly associated with OS. Furthermore, for RFS, CD8+ T cells (HR = 0.98) and M0 macrophages (HR = 1.02) were potential biomarkers in HBV−HCC, whereas neutrophils (HR = 1.01) were an independent predictor in HCV−HCC. Lastly, in both HBV−HCC and HCV−HCC, CD8+ T cells (HR = 0.97) and activated dendritic cells (HR = 1.09) had a significant association with OS, while γ delta T cells (HR = 1.04), monocytes (HR = 1.05), M0 macrophages (HR = 1.04), M1 macrophages (HR = 1.02), and activated dendritic cells (HR = 1.15) were highly associated with RFS. **Conclusions:** These findings demonstrated that TILs are potential survival predictors in HCC and different kinds of TILs are observed according to the virus type. Therefore, further investigations are warranted to elucidate the role of TILs in HCC, which may improve immunotherapy outcomes.

Hepatocellular carcinoma (HCC) is the most frequent liver malignancy and ranks as the second leading cause of cancer deaths in the world [1]. Although the majority of liver malignancy is related to viral hepatitis B (HBV) or C (HCV) [2], there are many differences between the two types of HCC in terms of activated pathways [3], gene expression profiles [4], immunologic responses [5], and clinical prognosis [6]. Recently, a meta-analysis of the effects of a targeted cancer drug, sorafenib, was conducted to evaluate whether there is a difference in overall survival (OS) between HBV− and HCV−induced HCC patients after receiving this drug [7]. This study discovered that the OS of HBV(+)HCV− patients after sorafenib treatment was significantly improved. In contrast, there was insufficient evidence to support a similar outcome in HBV(−)HCV(+) patients. Therefore, specific treatment strategies should be developed to treat virus-driven HCCs effectively.

Immune checkpoint therapy is widely used to treat melanoma, such as squamous-cell lung carcinoma [8], renal cell carcinoma [9], and bladder cancer [10]. However, local inhibition of the anti-tumor immune responses in the microenvironment makes immunotherapy challenging to implement [11]. Tumor-infiltrating leukocytes (TILs) are white blood cells surrounding the tumor stroma and inside the tumor [12]. TILs have a vital role in the prognosis and prediction of many malignancies [13,14], and their prognostic association with HCC has been widely investigated [15,16]. For example, the higher amount of macrophages discovered in HCC patients was associated with poor clinical outcomes [17,18] and the proportions of different kinds of macrophages (M1 and M2) also affected the prognosis [19]. In other studies, HCC patients who had more dendritic cells [20,21,22], natural killer cells [23], T lymphocytes [24,25] or B cells [26,27,28] had a better prognosis; while those who had more neutrophils [29,30], monocytes [31,32], regulatory T lymphocytes [33], and CXCR3+ subtype B cells [34] had the poorest prognosis. Hence, the understanding of the role of TILs in tumor immunology may help to overcome current barriers in anti-cancer treatment.

The differential gene expression patterns between HBV− and HCV−induced HCC have been investigated [35]. Norio Iizuka et al. found 31 differentially expressed genes (DEGs) in HBV−HCC that were involved in signal transduction, transcription, and metastasis, while 52 DEGs significantly increased in HCV−HCC were related to the immune system and detoxification [4]. This study concluded that the pathogenesis of HCC triggered by HBV versus HCV might be different. Moreover, the DEGs could be critical diagnostic markers, implying that distinct treatments may be applied to HBV−HCC and HCV−HCC patients.

Many studies have investigated TIL composition in HCC, the majority emphasizes HCC in general [36,37], individual HBV−/HCV−infected cases [38,39], or specific immune cell type [40]. However, few studies have focused on HBV−HCC versus non-viral HCC [41]. Also, very few studies have addressed issues such as the different immune response associated with HBV−HCC versus HCV−HCC. Therefore, this study aims to define the cell composition of the immune response in both HBV−HCC and HCV−HCC and to investigate its relationship with clinical outcomes such as OS and recurrence-free survival (RFS). We applied two well-established algorithms (ESTIMATE and CIBERSORT) to ascertain the relationship between the immune infiltrate composition and the clinical prognoses, hoping to arrive at a better understanding of virus-driven HCC.

## 1. Materials and Methods

### 1.1. Identification and Selection of Included Studies

To focus on TILs’ influence on HCC with different viral origins, we divided the HCC into three groups: HBV−related HCC, HCV−related HCC and unspecified HCC. Specifically, the HCC HBeAg-positive or HBcAb-positive samples were defined as HBV−related HCC, whereas sample testing positive by the anti-HCV agent were characterized as HCV−related HCC. To further identify the gene expression datasets in NCBI Gene Expression Omnibus (GEO) related to the three HCC types, we used the following search terms: “hepatitis B virus”, “hepatitis C virus”, “hepatocellular carcinoma”, “survival”, “clinical” and “recurrence”, which yielded 10 datasets. We also used a publicly available liver cancer dataset (*n* = 371) from the Cancer Genome Atlas (TCGA), which included both gene expression profiles and clinical information.

The primary inclusion criteria for the datasets in this study were: (1) viral status is clearly stated for HBV−HCC and HCV−HCC samples, and samples having no specific information are defined as HCC samples; (2) contains expression profiles of both tumor cells and the adjacent normal tissues for each sample; (3) includes at least one reliable clinical outcome such as OS rate; (4) is the most recently published version, or the version with the largest sample size when the same dataset is used repeatedly in different studies. However, some datasets were excluded for the following reasons: (1) they contained a mixture of two HCC types; (2) HCC was caused by metastasis; (3) the sample size was smaller than 30; (4) array/NGS platforms used in the dataset were not suitable for our downstream pipeline. These selection criteria yielded six datasets (GSE10143, GSE76427, GSE54236, GSE14520, GSE14520, and TCGA) for use in the analyses.

### 1.2. Statistical Analysis

The analysis pipeline used in this study is illustrated in Figure 1. The total TIL quantity and the proportion of immune cells in the tumor tissue or adjacent normal tissue from the same patient were calculated based on the expression profiles by ESTIMATE [42] and CIBERSORT [43] separately. The ESTIMATE algorithm uses gene expression data (tumor/non-tumor) to infer the percent composition of infiltrating immune cells in tumor tissues and is based on the calculation of the single sample Gene Set Enrichment Analysis (ssGSEA). For the ssGSEA algorithm, 171 marker genes are required to generate an immune score. This score is in proportion to the real quantity of immune cells; as a result, the number of immune cells among samples can be directly compared using this value. The CIBERSORT algorithm is a tool for estimating the abundances of immune cells using transcriptomic data. This software applies linear support vector regression (SVR) to deconvolve the relative fractions of immune cells from the transcriptional profiles of a bulk tumor sample based on the signature matrix as reference.

To handle the different batches and/or platforms used in the published datasets, the pipelines of the ESTIMATE algorithm and the CIBERSORT algorithm contained a normalization step. In order to minimize batch effects, we used quantile normalization in both the RNA-Seq expression data and microarray data within one dataset. For multiple datasets, the CIBERSORT algorithm reports the proportions of different TILs within each dataset, which means the proportions have already been normalized within each dataset and their sum equals 100. Therefore, it is feasible to use the proportions from different datasets to make the comparisons. However, for the ESTIMATE algorithm, different datasets may result in different values of the immune score and thus we compared the immune scores within each dataset instead of making the comparisons across different datasets.

The paired Wilcoxon signed-rank test was applied to identify the difference in abundance and types of TILs according to the virus type. *p*-values were corrected for multiple testing by the Benjamin-Hochberg method (known as *q*-value) [44]. To evaluate whether immune cells are associated with OS, a Cox proportional hazards regression model was used (the coxph function in the survival package of R). The statistical results (Cox coefficients, hazard ratios (HR) with 95% confidence interval (CI), and *p*-values) for each immune cell type were obtained. We used a univariate Cox regression model to determine the effect of clinical factors on OS. Additionally, the associations of each immune cell type and significant clinical factors with OS and RFS were assessed via multivariate Cox regression models using the *stats* R package.

## 2. Results

### 2.1. Selection of Included Datasets

The characteristics of the selected datasets are shown in Table 1. The patients in these datasets were from Japan, Singapore, China, Italy, and the United States, and all the data were collected in 2008 or later. Two datasets (GSE10143 and TCGA) included all three types of patients (HBV−HCC, HCV−HCC, and HCC), and four (GSE76427, GSE54236, GSE14520 and GSE17856) were restricted to at least one HCC type. In total, 1,276 patients’ samples were included in this study: 313 were HBV−HCC patients, 135 were HCV−HCC patients, and 880 were HCC patients. The median follow-up ranged from 0.99 to 7.8 years.

### 2.2. Estimation of Infiltrating Cells

ESTIMATE was used to compare the quantities of immune cells between intra-tumor and non-tumor samples from the same patient. In the five GEO datasets (GSE10143, GSE76427, GSE54236, GSE14520, and GSE17856) used in the estimation of the number of immune cells, the immune scores were significantly lower in intra-tumor samples (Figure 2). The *p*-values of each Wilcoxon signed-rank test were lower than 0.0001.

### 2.3. Composition of TILs

To systematically investigate the difference TIL composition between intra-tumor and non-tumor samples from the same HCC patient, we then applied CIBERSORT to calculate the proportion of TILs in each tissue type.

With respect to the HBV−HCC group (GSE14520, *n* = 204), the signatures of resting memory CD4+ T cells (*q*-value < 0.001), activated memory CD4+ T cells (*q*-value < 0.001), activated natural killer cells (*q*-value < 0.001), resting dendritic cells (*q*-value < 0.001), and resting mast cells (*q*-value < 0.001) were found to predominate in tumor tissue. In contrast, plasma cells (*q*-value = 0.001), CD8+ T cells (*q*-value < 0.001), γ delta T cells (*q*-value < 0.001), M1 macrophages (*q*-value < 0.001), M2 macrophages (*q*-value < 0.001), and activated mast cells (*q*-value < 0.001) highly dominated in non-tumor tissue (Figure 3, left and Table 2).

In the HCV−HCC group, two datasets (GSE10143, *n* = 46; GSE17856, *n* = 40) were investigated (Appendix A). In the GSE10143 dataset, the proportions of M0 macrophage and neutrophil cells were higher in the intra-tumor tissue (2.9% ± 3.8% and 5.1% ± 3.4%, respectively) than in non-tumor tissue (1.2% ± 2% and 3.4% ± 2.3%, respectively). In contrast, relatively higher numbers of both memory B cells and CD8+ T cells were present in non-tumor tissue. In the GSE17856 dataset, γ delta T cells (*q*-value = 0.05) and M0 macrophages (*q*-value = 0.002) were more abundant in the intra-tumor tissue while more monocytes (*q*-value < 0.001) were present in non-tumor tissue. After enlarging the sample size by combining these datasets (GSE10143 and GSE17856, *n* = 86) to increase the statistical power, the results revealed that there was a significantly larger proportion of M0 macrophages (*q*-value < 0.001) and neutrophil cells (*q*-value = 0.028) in the intra-tumor tissue; however, there were higher percentages of CD8+ cells (*q*-value = 0.04) and monocytes (*q*-value < 0.001) in non-tumor tissue (Figure 3, center).

With respect to the unspecified HCC group, either individual datasets or a pooled dataset (GSE74627, *n* = 52; GSE54236, *n* = 78; GSE10143, *n* = 62; GSE14520, *n* = 204; GSE17584, *n* = 40; total *n* = 437) were used for examination (Figure 3 and Appendix A). The results for the pooled dataset demonstrated that there were five immune cell types significantly more abundant in the intra-tumor tissue: CD4+ memory activated T cells (*q*-value = 0.038), NK activated cells (*q*-value < 0.001), M0 macrophages (*q*-value < 0.001), activated dendritic cells (*q*-value < 0.001), and resting mast cells (*q*-value < 0.001); whereas the immune cells detected more in non-tumor tissue were as follows: plasma cells (*q*-value < 0.001), CD8+ activated T cells (*q*-value < 0.001), γ delta T cells (*q*-value < 0.001), monocytes (*q*-value = 0.005), M1 macrophages (*q*-value < 0.001), M2 macrophages (*q*-value < 0.003), and mast activated cells (*q*-value < 0.001 (Figure 3, right). The TILs with the same infiltration pattern among three HCC groups are further summarized in Appendix A. This analysis showed that M0 macrophages were consistently more abundant in tumor tissue across HCC subtypes, while CD8+ T cells were more abundant in non-tumor tissue. Taken together, these results suggest that immune cells in different virus-driven HCC groups might play an important role in the process of carcinogenesis.

### 2.4. Prognostic Associations of Clinical Diagnoses and Immune Cells in Tumor Tissue

To determine the prognostic effect of immune cells in tumor tissue, we utilized OS and RFS as the indicators.

In the HBV−HCC group, two datasets (GSE14520, *n* = 204; TCGA, *n* = 95) were analyzed using univariate, adjusted univariate, and multivariate analysis (Appendix A). The clinical factors that affected patients’ OS and RFS in these two datasets are shown in Appendix A. In GSE14520, there were five clinical factors affecting patients’ OS: TNM stage, tumor size, AFP concentration and multinodular characteristic; whereas the factors significantly associated with patients’ RFS were as follows: gender, TNM stage, and BCLC stage. After adjusting for the above factors, activated dendritic cells (HR = 1.06; *p* = 0.01) and resting dendritic cells (HR = 1.09; *p* = 0.042) positively affected the OS of patients in this dataset. The univariate Cox regression of OS using a pooled dataset (GSE14520 and TCGA; total *n* = 299) revealed that plasma cells (HR = 1.06; *p* = 0.005) and activated dendritic cells (HR = 1.12; *p* = 0.006) positively contributed to the model, while M1 macrophages (HR = 0.96; *p* = 0.021) negatively contributed to the model. However, there was a slight difference when we excluded non-Asian patients (*n* = 287). According to the univariate analysis, plasma cells (HR = 1.06; *p* = 0.008), activated dendritic cells (HR = 1.04; *p* = 0.05), and resting dendritic cells (HR = 1.12; *p* = 0.007) were positively associated with OS; whereas M1 macrophages (HR = 0.96; *p* = 0.03) were negatively associated with OS in Asian patients. As for RFS, M0 macrophages (HR = 1.02; *p* = 0.005) had a negative effect but M1 macrophages (HR = 0.96; *p* = 0.033) had a positive effect.

Regarding the HCV−HCC group, three datasets (GSE10143, GSE17856, and TCGA) were investigated (Appendix A) using only univariate analysis, because there was either no clinical annotation or no significant association between known clinical factors and OS [45]. The unadjusted univariate analysis of a pooled dataset (GSE10143 and TCGA; total *n* = 95) revealed that lymphocytes positively associated with OS were resting NK cells (HR = 1.13; *p* = 0.021) and monocytes (HR = 1.21; *p* = 0.012). Another unadjusted Cox regression analysis of a pooled dataset (GSE10143 and GSE17856; *n* = 89) showed lymphocytes positively associated with RSF were resting NK cells (HR = 1.11; *p* = 0.035), M2 macrophages (HR = 1.04; *p* = 0.003), and neutrophils (HR = 1.10; *p* = 0.014).

For the HCC group, four datasets (GSE10143, *n* = 62; GSE76427, *n* = 52; GSE542365, *n* = 78; TCGA, *n* = 329) were individually analyzed using univariate, adjusted univariate, and multivariate analysis (Appendix A). The clinical factors that affected patients’ OS and RFS in GSE76427 are shown in Appendix A. The adjusted univariate analysis of a pooled dataset (GSE10143, GSE76427, GSE54236, GSE17856, GSE14520, TCGA; total *n* = 793) showed that CD8+ T cells (HR = 0.97; *p* = 0.015) were negatively associated with OS. In contrast, activated dendritic cells (HR = 1.09; *p* = 0.01) were positively associated with OS. Moreover, another pooled dataset (GSE10143, GSE76427, GSE14520, GSE17856; total *n* = 418) showed that seven immune cell types (plasma cells, γ delta T cells, resting NK cells, monocytes, M0 macrophage, M2 macrophage and activated dendritic cells) have a positive impact on RFS.

We further asked whether the same immune cells are associated with OS and/or RFS among the three HCC subtypes. Appendix A shows that RFS was negatively affected by neutrophils in both HBV−HCC (HR = 1.11) and HCV−HCC patients (HR = 1.20). Moreover, in HBV−HCC and HCC tumors, OS had a positive association with activated dendritic cells, and RFS showed a similar trend for M0 macrophages. Meanwhile, the proportion of resting NK cells and M2 macrophages in the HCV−HCC and HCC groups had a positive impact on RFS. Unfortunately, no common immune cells had a similar impact across all three HCC groups. Also, the TILs that both affected patient survival and were differentially expressed in the tumor cells are summarized in Table 3. For instance, plasma cells (HR = 1.05), M1 macrophages (HR = 0.95), and activated dendritic cells (HR = 1.08) were predictive of survival in HBV−HCC, whereas monocytes (HR = 1.21) predicted survival in HCV−HCC. These results suggest that the TILs serving as the survival predictors in HCC may vary based on the HCC subtype.

## 3. Discussion

### 3.1. Different Immune Responses in Different Subtypes of HCC

HBV is a small DNA virus with the ability to insert into the host genome, causing tumorigenesis. HBV generates regulatory protein HBx, which is involved in cell growth and carcinogenesis. In HCC, this protein has been reported to regulate the Wnt pathway [46] and interfere with the function of innate immunity [47]. In contrast, HCV is a positive-stranded RNA virus, which produces a nuclear protein that inhibits NK cells’ ability [48]. Therefore, different viruses might induce different TILs. To our knowledge, the distributions and proportions of TILs in HBV and HCV have not been comprehensively investigated.

Our results demonstrated that the immune infiltrates associated with OS in HCV−HCC patients were different from those found in HBV−HCC and HCC groups. Consistent with previous studies [49,50], this evidence proved that different virus status activated diverse immune responses that ultimately influenced the patient’s survival. However, because of the small sample size, there were also limitations associated with the statistical analysis. Regarding the HBV−HCC and HCC groups, the results revealed that both plasma cells and activated dendritic cells were negatively associated with OS. Dong-Ming Kuang pointed out that plasma cells secrete IgG to enhance M2 polarization, and M2-polarized macrophages push plasma cells to secrete more IgG, forming a positive-feedback loop in hepatoma [51]. Additionally, other studies indicated that environmental semimature dendritic cells might activate extra FcγRIIlow/− B cells in HCC tumors to suppress cytotoxic T cell function [52], and these semimature dendritic cells also induce immune tolerance by enhancing the production of regulatory T cells [53]. Therefore, our results from HBV−HCC and HCC cases are in line with those of previous studies.

We also found that the relative proportions of plasma cells, M0 macrophages, M1 macrophages, CD8+ T cells, and activated dendritic cells significantly varied between intra-tumor and non-tumor tissue in HBV−HCC samples; at the same time, these differences were also directly related to OS. Neutrophils and monocytes were predictive of survival in the HCV−HCC group, while CD8+ T cells and activated dendritic cells were predictive in the HCC group. These results indicate that immune infiltrates may participate in the process of oncogenesis, and the fact that their relative abundance differentially affected the survival time in each patient suggests that these immune cells may be potential targets in immunotherapy. For example, in HBV−HCC, corruption of dendritic cells or enhancement of M1 macrophages’ proliferation and function could act as anti-cancer strategies. Besides, both activated dendritic cells and M0 macrophages were more abundant in HBV−HCC and HCC, indicating that the oncogenic mechanism of non-viral HCC might be similar to that of HBV−HCC.

### 3.2. Limitations and Future Prospects

The prevalence of HCC in many Asian countries is generally higher than in western countries [54]. Infection with HBV is the leading risk factor for HCC in East and South-East Asian countries like China, South Korea, and Malaysia [55], whereas the incidence rate of HCV−HCC is high in other Asian countries like Japan and Singapore [56]. In this study, most of the available datasets with relatively larger sample size were also from Asia. The ratios of Asian samples to American samples in both HBV−HCC and HCV−HCC were around 2 to 1, whereas the ratio of Asian samples to other populations (European and American) in HCC was about 1.7:1. Therefore, it was not feasible to conduct a population-specific analysis by dividing the samples into different populations.

Few gene expression datasets of HCCs are currently available in the public databases, and our exclusion criteria further restricted the number of analyzable datasets. In addition, since all datasets were retrieved from the public domain, we can only analyze the variables provided in each dataset. Such limited clinical information on, for example, the follow-up time, might affect the analytical results. Therefore, further investigations are required and warranted to validate the results of this study.

Since both the total quantity and relative composition of TILs were different from cancer to cancer, the cytokines and the microenvironment centering on the TILs must also be different. Hence, it is possible that different TILs were observed in the adjacent normal tissue, even in the same organ. Furthermore, tumor purity varies across different samples, and thus, this might be another reason why we observed different TILs in the non-tumor cells. Lastly, although the adjacent normal cells were defined as non-tumor tissue, it is difficult to ensure that no lesion or tumor cells exist in them.

Previous studies have shown that Tregs accumulate in tumors and are associated with a poor prognosis for HCC [57]. However, the log-rank test revealed that the expression of Tregs was an effective predictor for only HCV−HCC (Appendix A). This may suggest that the different compositions of HCC types in a cohort lead to different prediction values of Tregs. If one cohort has more patients with HCV−HCC, Tregs expression might then be a possible predictor for prognosis. In addition, low detections of Tregs in the selected datasets may lead to unstable predictions. It is well-known that transcriptome analyses using bulk RNA-seq or transitional microarray may be difficult to capture low abundant or rare cell populations within the tumor; however, the latest single-cell RNA-seq analysis has been able to informatively and robustly investigate those lowly represented cell subsets in tumor [58]. Therefore, conducting analysis at the single-cell resolution may shed light on the role of the immune responses in HCC.

Lastly, the algorithms applied in this study, which depend on gene expression data rather than cell counts, might not be the most appropriate ones to identify which immune infiltrates are present, or to evaluate the differences in each TIL type. Although many studies still make use of relative expression values in survival prediction [41,59], future work should be undertaken to improve the ESTIMATE algorithm to compute an absolute score which is not affected by technical issues such as different platforms.

## 4. Conclusions

In this study, we have shown that the abundance of infiltrating leukocytes in non-tumor tissue is greater than in the intra-tumor tissue, and the relative composition of TILs among HBV−HCC, HCV−HCC, and HCC samples is diverse. Additionally, our results also revealed that different HCC groups had different immune cells affecting the OS and RFS of the patients. However, the robustness of results may be restricted to the small sample size and the transcriptomic analyses of bulk tumor cell populations. Therefore, further validations using a larger number of samples generating from advanced technologies are required, and future improvements in the prediction algorithms should also be applied to minimize the platform effects and explore the absolute quantity of each immune cell in non-tumor versus tumor tissue.

## Figures and Tables

**Figure 1 genes-10-00630-f001:**
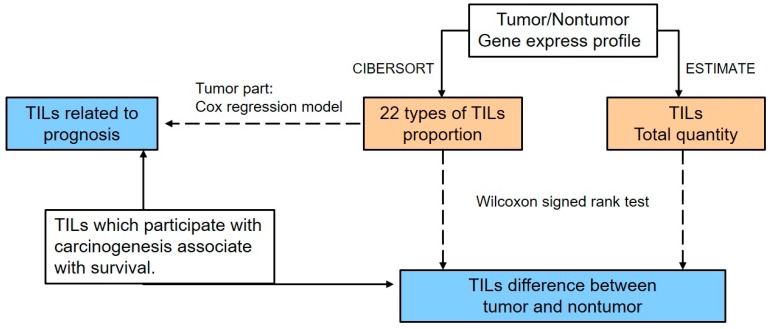
An overview of this study.

**Figure 2 genes-10-00630-f002:**
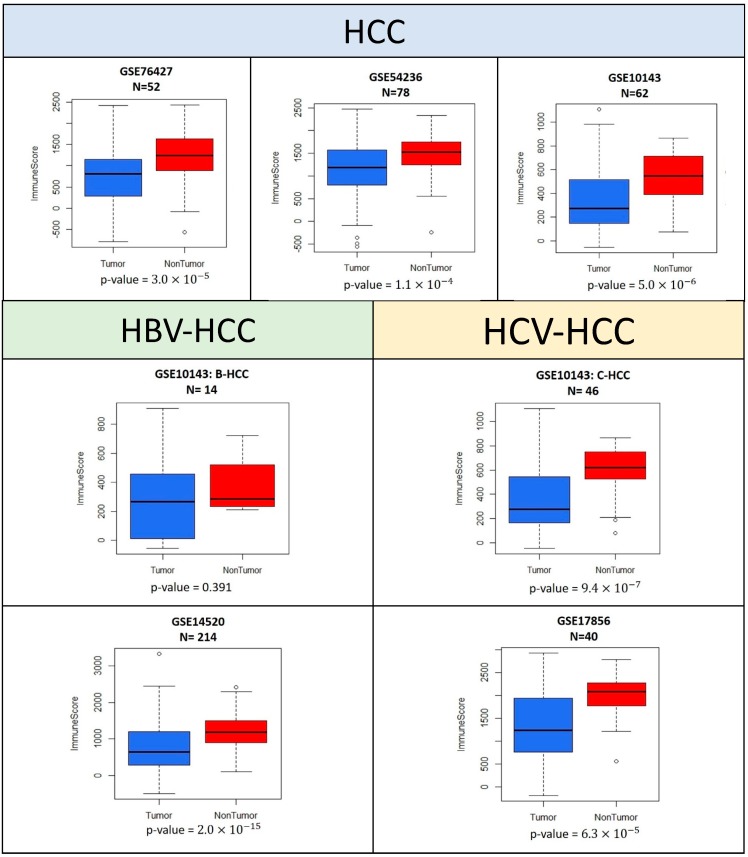
Immune scores for intra-tumor and non-tumor samples from hepatocellular carcinoma patients. The five datasets (GSE76427, GSE54236, GSE10143, GSE14520 and GSE17856) with paired data were quantile normalized and the normalized data were then fed into ESTIMATE software for the calculation of immune scores. *p*-values were obtained by Wilcoxon signed-rank tests.

**Figure 3 genes-10-00630-f003:**
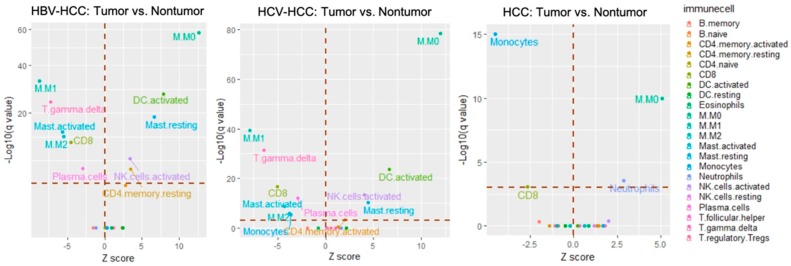
The relative composition of TILs in the intra-tumor and non-tumor tissue using CIBERSORT. The datasets used for these visual outputs were (left) HBV−HCC: GSE14520; (center) HCV−HCC: GSE17856 and GSE10143; (right) HCC: GSE74627, GSE54236, GSE10143, GSE14520, and GSE17584. All immune cell types were evaluated, but only the statistically significant ones are labeled on the plot. *p*-value was calculated by a Wilcoxon rank-sum test and then adjusted by Bonferroni correction (*q*-value). The red dotted line on the *y*-axis indicates a *q*-value of 0.05 (>0.05: non-significant; ≤0.05: significant) whereas the one on the *x*-axis indicates a Z score of 0 (>0: relatively higher abundance in tumor tissue; <0: relatively higher abundance in non-tumor tissue).

**Table 1 genes-10-00630-t001:** Characteristics of 6 public datasets and the classifications of HCC types.

Datasets	Year	Country	Sex (M/F)	Mean Age	TNM Stage	No. of Patients		
						HBV−HCC *^a^*	HCV−HCC *^b^*	HCC ^c^
*Paired data*								
**GSE10143**	2008	Japan	-	-	-	14	46	62
**GSE76427**	2018	Singapore	45/7	59.5 ± 12.5	I: 28	-	-	52
					II: 12			
					III: 12			
**GSE54236**	2016	Italy	61/17	-	-	-	-	78
**GSE14520**	2010	China	178/26	50 ± 10.6	I: 88	204	-	204
					II: 74			
					III: 42			
**GSE17856**	2010	Japan	-	-	-	-	40	40
*Tumor-only data*								
**GSE76427**	2018	Singapore	93/22	63.5 ± 12.7	I: 55	-	-	115
					II: 35			
					III, IV: 24			
					NA: 1			
**TCGA**	2014	United States	107/222	55 ± 11.7	I: 161	95	49	329
					II: 82			
					III, IV: 86			
**Total**						313	135	880

*^a^* HBV−HCC: hepatitis B-related hepatocellular carcinoma; *^b^* HCV−HCC: hepatitis C-related hepatocellular carcinoma; *^c^* HCC: hepatocellular carcinoma caused by either a virus or other reasons.

**Table 2 genes-10-00630-t002:** The comparative composition of TILs in HBV−HCC.

GSE No.	Group	Tissue	Immune Scores	*p*- value	Cibersort		*p*- value *^c^*	*q*- value *^d^*
					Types	Mean of Proportion (Tumor/Nontumor) *^b^*		
GSE14520	HBV−HCC *^a^* (*n* = 204)	Tumor	782.7 ± 643.6	<0.001	T cells CD4 memory resting	6.4 ± 8.4/3.9 ± 6	<0.001	0.003
		Nontumor	1215.4 ± 425.9		T cells CD4 memory activated	0.4 ± 1.4/0.1 ± 0.6	<0.001	0.038
					NK cells activated	7.1 ± 4.4/5.7 ± 4	<0.001	<0.001
					Macrophages M0	12.1 ± 11.1/1.6 ± 4	<0.001	<0.001
					Dendritic cells activated	1.4 ± 2.3/0.1 ± 0.4	<0.001	<0.001
					Mast cells resting	3.8 ± 4.8/1.3 ± 2.4	<0.001	<0.001
					Plasma cells	4.7 ± 3.4/5.6 ± 2.8	<0.001	0.001
					T cells CD8	11.2 ± 8.5/14.8 ± 7.4	<0.001	<0.001
					T cells γ delta	5.7 ± 5.2/10 ± 6.5	<0.001	<0.001
					Macrophages M1	10.9 ± 5.7/15.7 ± 5.4	<0.001	<0.001
					Macrophages M2	9.7 ± 6.2/13.4 ± 7.5	<0.001	<0.001
					Mast cells activated	1.9 ± 3/4 ± 4.4	<0.001	<0.001

*^a^* HBV−HCC: hepatitis B-related hepatocellular carcinoma; *^b^* Proportion values (tumor/nontumor) were expressed as mean ± SEM; *^c^ p*-values were obtained by Wilcoxon signed-rank tests; *^d^* Benjamini-Hochberg adjusted *p*-values were denoted as *q*-values.

**Table 3 genes-10-00630-t003:** Summary of the TILs that both were differentially expressed in the tumor cells and affected the survival of the patient.

Groups	Overall Survival (HR *)	Recurrence-Free Survival (HR *)
HBV−HCC	Plasma cell (1.05) Macrophages M1 (0.95) Dendritic cells activated (1.08)	T cells CD8 (0.98) Macrophages M0 (1.02)
HCV−HCC	Monocytes (1.21)	Neutrophils (1.10)
HCC	T cells CD8 (0.97) Dendritic cells activated (1.09)	Plasma cells (1.05) T cells γ delta (1.04) Monocytes (1.05) Macrophages M0 (1.04) Macrophages M2 (1.02) Dendritic cells activated (1.15)

* HR: Hazard Ratio.

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
