# Peer review of "Tumor-Infiltrating Leukocyte Composition and Prognostic Power in Hepatitis B- and Hepatitis C-Related Hepatocellular Carcinomas"

_genes, 2019, doi:10.3390/genes10080630_

Round 1

Reviewer 1 Report

The manuscript by Hsiao et al. employed ESTOMATE and CIBERSORT on 6 public HCC datasets to interrogate the tumour-infiltrating leukocytes (TILs) in all HCC, HBV- or HCV-related HCC. The analysis provided herein is focused however more detailed information could be obtained if deeper analyses could be performed. More specifically:

1. As it involves 6 different databases, did the authors perform normalization of the data across these datasets to come to the conclusion? If so, please provide more details in material and methods.

2. Is Fig.3 based on a pooled dataset? If so, please state clearly which datasets were pooled in each analysis in Fig.3.

3. Have the authors tried comparing the TIL composition across tumours or the ratio of each TIL composition tumour/non-tumour from each HCC etiologies? This will allow a direct comparison between HCC from different etiologies rather than the current side by side comparison of each etiologies between their respective tumour versus non-tumour tissues.

4. It was previously shown that Treg accumulate in tumour and has a poor prognostic value for HCC. It is surprising that Treg was not present in either of the analyses herein. Did the authors have reliable and detectable Treg from the RNA seq data?

5. Whenever the TILs were mentioned to affect the survival of HCC patients, it was not always indicated if it is the higher or lower proportion of the TILs that affected the survival in positive or negative way. This has to be stated clearly.

6. It will be interesting if the authors could provide analysis on other clinical parameters such as tumour size, stage or grade of the tumour.

Minor points:

1. Reference is missing for first result for B-HCC group TIL composition analysis.

2. For conclusion, it is not true that TILs in HBV & HCV has not been investigated before; for instance, Lim et al. in Gut 2018 has investigated the differences in TILs composition between HBV and non-viral related HCC and have significant differences.

Author Response

Enclosed please find the response in the cover letter.

Reviewer 2 Report

Hsiao et al investigated the expression of TILs in HCC of different etiologies and their impact on OS and RFS, using bioinformatic algorithms.

It is an interesting and informative study with the limitations which are mentioned by the authors, the main limitation being the heterogeneity of the data used since the study is based in already published data.

Another problem is that in some parts the use of English language needs more refining: the important paragraph in page 3 lines 98-107 where the inclusion criteria are described needs re-writing in a more comprehensible format.

In addition, the two platforms used, ESTIMATE and CIBERSORT need some short description, because the reader can not understand the analysis if he is not familiar with the two references which describe the algorithms.

I could not identify the results of the multivariate analysis. Were immune scores independent predictors of OS and RFS in the multivariate analysis together with statistically significant clinical data?

Author Response

Enclosed please find the response in the cover letter

Round 2

Reviewer 1 Report

I can see the authors have made tremendous effort in improving the manuscript. Just a few comments remain:

Do acknowledge the limitation of detecting Treg in the dataset. It is generally known that Treg detection might be low or unreliable with RNA seq data as they can be a rarer subsets in the tumour tissues and hence may be hard to capture. Lim et al focus only on HBV HCC vs non viral HCC no HCV HCC was mentioned. Also do check the reference, from the clean version I read, the citation to this paper seems to be wrong. U may want to further improve on language e.g. check for errors in spelling, grammar and avoid generalized/sweeping conclusion based on limited data without further validation.

Author Response

Enclosed please find the response of each comment in the attached file.
